# De Novo Polycomb Recruitment: Lessons from Latent Herpesviruses

**DOI:** 10.3390/v13081470

**Published:** 2021-07-27

**Authors:** Sara A. Dochnal, Alison K. Francois, Anna R. Cliffe

**Affiliations:** Department of Microbiology, Immunology and Cancer Biology, University of Virginia, Charlottesville, VA 22903, USA; sd9fb@virginia.edu (S.A.D.); akf2xu@virginia.edu (A.K.F.)

**Keywords:** polycomb silencing, virus, herpesvirus, latency

## Abstract

The Human Herpesviruses persist in the form of a latent infection in specialized cell types. During latency, the herpesvirus genomes associate with cellular histone proteins and the viral lytic genes assemble into transcriptionally repressive heterochromatin. Although there is divergence in the nature of heterochromatin on latent herpesvirus genomes, in general, the genomes assemble into forms of heterochromatin that can convert to euchromatin to permit gene expression and therefore reactivation. This reversible form of heterochromatin is known as facultative heterochromatin and is most commonly characterized by polycomb silencing. Polycomb silencing is prevalent on the cellular genome and plays a role in developmentally regulated and imprinted genes, as well as X chromosome inactivation. As herpesviruses initially enter the cell in an un-chromatinized state, they provide an optimal system to study how de novo facultative heterochromatin is targeted to regions of DNA and how it contributes to silencing. Here, we describe how polycomb-mediated silencing potentially assembles onto herpesvirus genomes, synergizing what is known about herpesvirus latency with facultative heterochromatin targeting to the cellular genome. A greater understanding of polycomb silencing of herpesviruses will inform on the mechanism of persistence and reactivation of these pathogenic human viruses and provide clues regarding how de novo facultative heterochromatin forms on the cellular genome.

## 1. Introduction

Herpesviruses are double-stranded DNA viruses that persist for life in specialized cell types in the form of a latent infection. During latency, herpesvirus genomes are associated with cellular histone proteins, forming a chromatinized structure [1]. Transcriptionally silent regions of the herpesvirus genomes are associated with histones with post translational modifications (PTMs) characteristic of repressive heterochromatin [1,2,3,4]. By forming a heterochromatin structure, herpesvirus genomes likely avoid recognition as foreign pieces of DNA and permit long-term silencing of lytic genes, enabling them to go undetected by the host.

Differential histone PTMs are associated with different forms of heterochromatin and are laid down and removed by distinct cellular proteins. A common feature of many human herpesviruses is the association of latent genomes with polycomb group (PcG) protein-mediated facultative heterochromatin (fHC). fHC is characterized by di- and trimethylation of H3K27 (H3K27me2/3) and can be accompanied by histone H2AK119 mono-ubiquitination (H2AK119ub1) and the stable recruitment of PcG complexes [5]. On the host genome, the composition of fHC can vary on different regions of DNA and in different cell types [6,7,8]. The differential composition of fHC arises due to the action of discrete PcG-associated proteins and results in a distinct mechanism to maintain transcriptional repression [9,10]. Therefore, there is great interest in understanding the mechanisms by which regions of DNA are specified for polycomb-mediated fHC silencing. However, tracking de novo fHC targeting to regions of the cellular genome is challenging as the full removal of the pre-existing epigenetic template is hard to achieve [11,12].

Herpesvirus genomes packaged into the infectious viral particle are not associated with histones and are, therefore, described as being “naked” [13]. Therefore, when herpesvirus genomes first enter the cell, they do so without a pre-existing chromatin template. Investigating how herpesvirus genomes are recognized and targeted for fHC silencing serves as a tool to understand the mechanisms of de novo formation of fHC. In addition, an understanding of how herpesvirus genomes are silenced during a latent infection will uncover mechanisms of how herpesviruses genomes are silenced in a way that can permit gene expression to later occur for reactivation and dissemination. This review will focus on evidence of fHC-based silencing of latent herpesvirus genomes. In addition, we synergize recent developments in the mechanisms of fHC formation on both cellular and viral genomes and posit how herpesviruses can be utilized in the future to understand the mechanisms of fHC-based gene silencing.

## 2. Polycomb Group Protein Repressive Complexes

As its name suggests, PcG-mediated heterochromatin is laid down and maintained by the activities of two major protein-complexes: the Polycomb repressive complex 1 (PRC1) and PRC2. The names and abbreviations of Polycomb-associated proteins are shown in Table 1. PRC2 was long considered the initial depositor of fHC, as it contains H3K27me ‘writer’ activity, carrying out mono-, di- or tri-methylation (H3K27me1, H3K27me2, or H3K27me3) [14,15,16]. Because H3K27me3 can then be recognized by the PRC1 complex, the formation of PcG-mediated silencing was thought to occur hierarchically with PRC2-mediated H3K27me3 followed by the recruitment of PRC1 to result in H2A lysine 119 mono-ubiquitination (H2AK119ub1) and/or chromatin compaction [17]. Our understanding of mammalian polycomb repression has since evolved beyond this model and the order of events in polycomb silencing seems far from straightforward [18]. Mammalian polycomb silencing likely occurs through a variety of mechanisms specific to cellular context and results in potentially divergent association with protein complexes and mechanisms to restrict gene expression. It is likely that herpesviruses have evolved to take advantage of host processes of heterochromatin formation.

### 2.1. H3K27 Methylation by PRC2

PRC2 contains a core of three components: enhancer of Zeste 1 or 2 (EZH1/2), suppressor of Zeste 12 (SUZ12) and embryonic ectoderm development (EED) (see Figure 1). EZH1/2 is catalytically active, but the full trimeric core is required for in vitro H3K27 methylation activity [16,19,20]. PRC2 core proteins also include retinoblastoma-associated proteins 46 and 48 (RbAp46/48; also known as RBBP4/7). All three methylation states of H3K27 (mono, di and tri-methylation) are carried out by PRC2. The formation and function of K27 methylation has been most extensively studied in mouse embryonic stem cells (mESCs) and human-induced pluripotent stem cells (iPSCs), where H3K27me2 is most abundant of the three, existing across 50–70% of total histone H3, predominately at intergenic regions [11]. Despite being the most predominant form of H3K27 methylation and a substrate for PRC2, PRC2 levels are undetectable at regions of H3K27me2 [11]. H3K27me1 and H3K27me3 are each found on approximately 10–15% of total H3 in mESCs [12,21]. H3K27me3 is found centered around CpG islands of transcriptionally silent genes, and is the only methylation state to which PRC2 stably binds [11]. Hence, these CpG islands can act as nucleation sites for facultative heterochromatin formation and propagation [11].

In addition to the core proteins, PRC2 is also associated with a variety of accessory proteins, and the mutually exclusive binding patterns give rise to two distinct PRC2 variants: PRC2.1 and PRC2.2 [22] (Figure 1). The possible combinations of accessory proteins for a single PRC2 variant reflect the built-in complexity in regulating PRC2 activity. The PRC2.1 complex includes one of three polycomb-like (PCL) proteins (PCL1/2/3), also known as PHF1, MTF2 and PHF19, respectively, which enable PRC2.1 binding to unmethylated CpG islands [22]. In addition to PCL proteins, PRC2.1 contains Elongin BC and Polycomb repressive complex 2-associated protein (EPOP) or PRC2-associated LCOR isoform 1 (PALI1/2) [23,24,25,26]. In contrast, PRC2.2 contains Jumonji and AT-rich interaction domain 2 (JARID2), and adipocyte enhancer-binding protein 2 (AEBP2), which together recruit PRC2.2 to the H2AK119ub1 modification and permit the PRC2 methyltransferase activity to overcome the repressive effects of active histone modifications H3K4me3 and H3K36me3 [27,28,29,30]. Although the differential activities of PHF1/PHF19/MTF2 and JARID2 contribute to the recruitment of PRC2.1 and 2.2 to different regions of chromatin, there are also redundant or compensatory mechanisms observed for PRC2.1 and PRC2.2 and inhibition of both complexes is required for the full inhibition of PRC2 activity in pluripotent cells [31]. It should also be noted that studies of PRC2.1/2.2 function on cellular chromatin have been carried out with some residual epigenetic template. In addition, it is likely that there are differences in the resulting epigenetic structures when fHC is laid down by PRC2.1 versus PRC2.2 that could result in different mechanisms for the re-expression of genes. This is also supported by a recent study that has uncovered subtle differences in PRC2.1 versus PRC2.2 by inducing mutations in SUZ12 to shift preference towards forming one complex over the other. This revealed that PRC2 occupancy was significantly higher for PRC2.1 target regions than PRC2.2 bound regions, suggesting that differential accessory proteins bound to either PRC2.1 or 2.2 can result in differing PRC2 occupancy levels, at least in human iPSCs [32]. Whether this has long-term implication for gene expression and chromatin compaction is not known. In addition, as gene expression patterns change during differentiation, there is evidence that PRC2.2 is important for de novo silencing during this transitional process, whereas MTF2-containing PRC2.1 was required to maintain high amounts of H3K27me3 at already repressed CpG dense promoters [7]. These results do not conflict with findings of redundancy between PRC2.1 and PRC2.2 in mESCs [33,34], instead suggesting that, during differentiation, their roles shift.

Hence, when considering the mechanisms of PRC2 recruitment and function, it is important to consider changes in Polycomb group expression and activity upon differentiation and in distinct cell types differentiated cell types, which is reflected in the differential composition of PRC2 in different cell types. One aspect that varies between pluripotent and differentiated cells is the presence of EZH1 versus EZH2. EZH2 is highly expressed in dividing cells and minimally in differentiated cells [35,36]. EZH1 is widely expressed and believed to be predominant in terminally differentiated cells, specifically in the maintenance of H3K27me3 [29]. Studies in mESCs indicate that EZH1/2 use different mechanisms to repress chromatin, with EZH2 largely catalyzing H3K27me2/3 and EZH1 contributing to chromatin compaction [35]. Different forms of Polycomb-associated proteins may also be present in distinct cell types. For example, full length JARID2 is down-regulated with differentiation, although a cleaved protein lacking the PRC2-interacting domain (DelN-JARID2) is present, at least in differentiated keratinocytes. The shorter form of JARID2 negatively regulates PRC2 function and is proposed to release PRC2-mediated repression of differentiation genes [37]. This serves as an example of the function of a polycomb protein changing in the context of differentiation, one facet of the complexities of polycomb repression that must be considered in heterochromatin formation. Models of herpesviruses latency and reactivation serve as excellent systems to investigate de novo fHC formation in differentiated cells, and also track the potential differential regulation of gene expression from regions of fHC within distinct epigenetic structures.

### 2.2. H2AK119 Ubiquitination, Chromatin Compaction and 3-Dimensional Interactions by PRC1

The composition of PRC1 complexes appear to be even more diverse than PRC2, falling into two broad categories: canonical (cPRC1) and non-canonical/variant (vPRC1) [9,10] (see Figure 2). All PRC1 complexes contain dimerized RING1A or B, which interact with one of six possible PCGF proteins (PCGF1-6). cPRC1 complexes contain either PCGF2 (also called MEL18) or PCGF4 (also called BMI1) together with one of five chromobox (CBX2, 4, 6, 7 and 8) proteins [38]. The CBX subunit of cPRC1 directly binds H3K27me3 for recruitment to chromatin; therefore, the activity of cPRC1 is dependent on PRC2 [39,40,41]. vPRC1 complexes can contain any of the six PCGF proteins but lack CBX proteins, instead containing either RYBP or YAF2 [18,38,42]. Hence, vPRC1 complexes lack H3K27me3 binding ability and are recruited to chromatin independently of PRC2 [18,43]. Recruitment of vPRC1 can occur through direct DNA binding activities; for example, KDM2B-containing vPRC1 directs the complex to non-methylated CpG islands [44]. vPRC1 can interact with sequence-specific DNA binding proteins such as E2F6, REST and RUNX1 (Figure 2) [45]. vPRC1 can also be recruited by interaction with RNA (See Section 4.4 below), as appears to be the case for vPRC1 recruitment to the presumptive inactive X-chromosome in mESCs by interaction of the non-coding RNA Xist via hnRNPK [46].

The variability in composition of PRC1 complexes is reflective of their differential effects on chromatin and mechanism to impact gene expression. Although all complexes contain the RING1/2 catalytic subunit, vPRC1 is predominantly responsible for mediating H2AK119ub1, likely because RYBP stimulates PRC1 E3 ubiquitin ligase activity [18,41,43,47]. There is evidence, at least in embryonic stem cells, that H2AK119ub1 is required for PRC2 recruitment and the subsequent binding of both vPRC1 and cPRC1 [48,49]. These studies also revealed a direct role for H2AK119ub1 in repression of gene expression, and there is evidence for H2AK119ub1 inhibiting RNA polymerase elongation [50,51], perhaps by preventing the eviction of H2A/H2B dimers [52].

Whilst vPRC1 is likely responsible for the majority of H2AK119ub1, cPRC1 appears to be responsible for repressing gene expression through chromatin compaction, in addition to mediating long-range chromosomal interactions [53]. For regions of chromatin that are bound by either vPRC1 or cPRC1 in ES cells, those bound by cPRC1 exhibit increased transcriptional repression compared to those bound by vPRC1 [54]. cPRC1 compaction is driven by the CBX proteins, in particular CBX2, 6 and 8, which contain a highly basic region that can drive chromatin compaction, at least in vitro [10]. Thus, CBX2, 6 and 8-containing cPRC1 likely prevents gene expression by limiting access to transcriptional machinery. CBX7, which is the most abundant CBX in ES cells, lacks this basic domain and instead may function in transcriptional repression via mediating long-range chromosomal interactions [10,54,55]. Other CBX proteins can also mediate long-range chromosomal interactions. Notably, CBX2 contains an intrinsically disordered region, which drives the formation of nuclear condensates known as polycomb group bodies by phase separation [56]. Therefore, differential expression of CBX paralogs in different cell types can lead to different degrees of compaction, the significance of which in herpesvirus latency remains to be resolved.

In mESCs, enzymatic activity of RING1A/1B is required for H2AK119ub1 and the majority of H3K27me3, indicating that at least in stem cells vPRC1-mediated polycomb repression is the predominant pathway [48,49]. Inhibition of this pathway also results in decreased association of cPRC1 at target genes usually bound by both vPRC1 and cPRC1 [48,49]. However, in these studies, small subsets of genes were found to be associated with H3K27me3 and cPRC1 via PRC2.1-mediated H3K27me3 deposition. There is evidence for a pathway of vPRC1-PRC2.2-mediated fHC formation during early development of neural progenitor cells (NPCs), where the RING1A/B E3 ligase activity was found to be essential for the repression of genes initially silenced during early neuronal development, but PRC1-mediated repression switched over to Ub-independence as stable silencing was maintained [57]. This result highlights the importance of considering cell type and differentiation state when exploring mechanisms of polycomb repression on herpesvirus genomes.

## 3. Evidence for Polycomb Group Silencing of Latent Herpesvirus Genomes

Herpesviruses are divided into three groups: the Alpha, Beta and Gammaherpesviruses. All establish a life-long latent infection with the capacity to undergo reactivation, but with different requirements depending on the nature of the latently infected cell. The Alphaherpesviruses establish latency in terminally differentiated neurons, most often in sensory and autonomic neurons [58]. The Betaherpesviruses persist in myeloid/dendritic progenitor cells, and the prototypical human Betaherpesvirus Human Cytomegalovirus (HCMV) establishes latency in hematopoietic lineage cells, including CD34+ progenitor stem cells, CD33+ granulocyte/macrophage progenitor cells, and CD14+ monocytes [59,60,61]. Gammaherpesviruses establish latency within lymphocytes, in addition to other cells types including epithelial and natural killer cells for Epstein–Barr Virus (EBV), and endothelial cells for Kaposi’s Sarcoma-Associated Herpesvirus (KSHV) [62,63,64,65]. For the majority of herpesviruses, latent viral genomes persist as circular, extrachromosomal episomes/plasmids, with the exception being the Betaherpesvirus HHV6, which has the ability to integrate into telomeric regions of host chromosomes [66].

When Herpesvirus genomes first enter the nucleus, they are not associated with cellular histone proteins and are, therefore, epigenetically naïve [67,68,69,70,71,72]. However, as latency is ultimately established, viral genomes become chromatinized by association with cellular histone proteins [1,2,73,74]. Non-nucleosome associated DNA in the nucleus can serve as a danger signal, potentially alerting the host cell to the presence of viral infection. Therefore, association with cellular histones likely protects the viral genome from recognition of the host innate immune response and thereby contributes to genome persistence [75,76,77,78,79,80,81]. Between the different herpesviruses, and even for the same herpesviruses, there is evidence of association with different types of heterochromatin characterized by the association with histone modifications found on regions of constitutive and facultative heterochromatin, with an example being HSV-1 as both H3K27me3 and H3K9me2/3 have been found on the latent viral genome [82,83,84]. However, to a variety of degrees, the majority of latent herpesvirus genomes investigated so far associate with PcG protein-mediated fHC. Given that by definition fHC is thought have the ability to convert to active euchromatin, particularly in response to stimuli that trigger cellular differentiation or stress signaling, it is perhaps not surprising that latent herpesvirus genomes are associated with this reversible form of heterochromatin.

The evidence linking the association of fHC with herpesvirus latency includes mechanistic studies investigating the role of K27me histone readers, writers, and erasers in regulating viral gene expression during latency/reactivation, in addition to studies demonstrating the association of proteins and histone modifications with latent viral genomes. Assessing the contribution of fHC to herpesvirus latency is, of course, dependent on the choice of experimental system used. For some herpesviruses, the epigenetic structure of latent genomes has been assessed in material isolated from humans to aid in our understanding of natural latency. This has been achieved for the Gammaherpesviruses and HCMV [85,86,87,88]. Passaged patient-derived cell lines containing latent KSHV and EBV have been incredibly powerful for understanding the molecular aspects of Gammaherpesvirus latency, including the role of fHC [89,90,91,92]. For the Alphaherpesviruses, very few studies have been performed to investigate the epigenome during natural latency due to the obvious challenge of removing neurons from healthy individuals, and autopsy studies are challenging as viral genomes need to be maintained in the natural latent state. Given the hurdles in investigating the HSV chromatin structure in natural latency, model systems are required. These are also advantageous as the mechanisms of fHC targeting during the establishment of latent infection can be more readily studied. Small animal models of herpesvirus latency have been invaluable for the initial characterization of the nature of the chromatin on the HSV genome [82,83,84,93,94,95]. However, the remaining human herpesviruses do not readily infect mice, although natural herpesviruses of small animals (for example, the wood mouse herpesvirus MHV-68/MuHV-4) have provided model systems to investigate chromatin association with latent genomes [96].

Some of the most comprehensive studies of fHC association with latent viral genomes have been performed using in vitro systems of Gammaherpesvirus latency, largely owing to the ability of both Epstein–Barr Virus (EBV) and Kaposi’s Sarcoma-Associated Herpesvirus (KSHV) to readily enter a latent state following infection of endothelial or renal carcinoma cells in particular. In vitro model systems based on primary cells isolated from humans or rodents are also used to investigate the epigenetics HCMV and HSV latency [85,97,98]. To generate larger amounts of cells and/or model latency in humans, immortalized cell lines or pluripotent cells can also be differentiated into the appropriate cell type [99,100]. Although more readily manipulated, in vitro systems come with the caveat that infection is carried out in isolated cells in the absence of a full host immune response. In addition, care needs to be taken when assessing the starting material; for example, whether the cells are embryonic or fetal in nature and may not fully mimic the chromatin environment in more mature cells [101,102]. Likewise, species-specific differences may underlie the chromatin responses to herpesvirus infection. Hence, it is advantageous to investigate the chromatin structure of herpesvirus genomes in multiple model systems. The availability of multiple systems has both drawbacks and advantages, as we can amass knowledge on the overall mechanisms of polycomb targeting in different models of herpesvirus infection in addition to how this may vary between different cell types, species, developmental stages, and the contribution of the host immune response.

### 3.1. Polycomb Group Silencing of Human Alphaherpesviruses

Using mouse models of HSV-1 latency, multiple studies have found that the viral genome is highly enriched for H3K27me3 [82,84,103]. The only region of the HSV genome expressed to abundant levels during latency is the latency-associated transcript (LAT) locus [1]. The LATs are a family of HSV non-coding RNAs expressed from the same region of the genome, the most abundant species being a stable approximately 2 kb intron, in addition to miRNAs and other long non-coding RNAs [104]. Surprisingly, the LAT promoter is also enriched for H3K27me3 during latency, but a downstream region known as the long-term enhancer appears to be depleted for H3K27me3 association [82]. The contribution of H3K27me3 to HSV latency is also demonstrated by the observed requirement for the activity of JMJD3 and/or UTX, H3K27 demethylases, in HSV-1 reactivation [97,105]. Consistent with H3K27me3 association with latent HSV, SUZ12 is associated with the HSV-1 genome as latency is established [95]. Recruitment of the PRC1 complex to the latent HSV-1 genome has been analyzed by ChIP assay for PCGF4/Bmi1, and no significant enrichment was detected on the lytic promoters examined [84,95]. PCGF4 can be a subunit in either vPRC1 or cPRC1, although there are multiple PRC1 variants that lack PCGF4, as discussed above. Therefore, it remains to be determined whether non-PCGF4 variant or canonical PRC1 proteins are present on the latent HSV-1 genome.

Although there are significant data showing that PcG-mediated fHC is enriched on the HSV-1 genome during latency, there is less known about the remaining human Alphaherpesviruses. There are no reported studies on the epigenetic nature of the HSV-2 genome during latency. Although there is approximately 50% sequence homology between HSV-1 and HSV-2, there are notable differences between the two viruses in terms of ability to undergo reactivation and express the LAT from different subtypes of neurons and anatomical locations [106,107]. Whether this relates to differences in the chromatin structure of HSV-1 versus HSV-2 is not known. As of yet, there are no data on the association of the varicella-zoster virus (VZV) latent genome with H3K27 methylation. The lack of small animal models that permit the establishment of latent infection with VZV has made the study of VZV latency more challenging. ChIP assays would be possible on human autopsy material, given the assumption that tissue can be processed rapidly and/or a significant proportion of viral genomes remain in the latent state. With the advent of more in vitro systems to investigate VZV latency in neurons generated from iPS cells [108,109], this opens the door to investigate the contribution of fHC to VZV latency.

### 3.2. Polycomb Group Silencing of Human Betaherpesviruses

For the prototypical human Betaherpesvirus, human cytomegalovirus (HCMV), there are no studies reported to date on the association of the HCMV genome with fHC during natural latency of humans. However, there is evidence that fHC chromatin plays a role in the maintenance or establishment of HCMV latency in experimental systems. In regard to the role of fHC in latency establishment, small molecule inhibition of PRC2 prevented quiescence in both human monocytic leukemia THP1 cell line models and embryonal carcinoma NT2D1 cell latency and reactivation models [110]. In addition, small molecule inhibition of PRC2 was sufficient to disrupt quiescence in NT2D1/THP1 models and knockdown of H3K27 demethylase JMJD3 reduced viral gene expression in a THP-1 reactivation model [110,111]. Further, a study by the Kalejta lab found that the H3K27 demethylases JMJD3 and UTX were targeted to the HCMV MIEP in both THP1 cells and CD34+ cells as a potential way for the host cell to defend against latency [112]. The MIEP is composed of the major immediate early promoter and enhancer regions and exerts control over immediate early gene expression, including IE1 and IE2, which are critical to initiating the viral cycle. In the presence of a viral protein UL138, recruitment of HDMs is decreased, permitting increased association of H3K27me3 with the MIEP. In a CD14+ experimental system of latency, overexpression of JMJD3/UTX resulted in increased lytic gene expression [98]. Together, these studies suggest that H3K27me3 association with the HMCV genome plays a role in latent infection. Moreover, it should also be noted that H3K9me3, along with the HP1 reader protein, is also known to be enriched on the HCMV MIEP during latency [113,114]. Hence, the contribution of Polycomb-mediated silencing and any potential intersection with HP1-mediated silencing in populations of latent HMCV genomes remains to be determined.

Betaherpesvirus HHV-6 is a remarkably silent virus; there are few if any detectable lytic or latent transcripts expressed in clinical isolates or in vitro infected cells [115,116]. A pioneering study recently found detectable levels of both H3K9me3 and H3K27me3 in iciHHV-6A cells, clonal cells derived from germ cells in which HHV-6 has integrated into the host cell [116]. There is currently no evidence for heterochromatin targeting the HHV-7 genome.

### 3.3. Polycomb Group Silencing of Human Gammaherpesviruses

For the Gammaherpesvirus Kaposi’s sarcoma-associated herpesvirus (KSHV), substantial evidence for polycomb-mediated silencing during latency has come from investigating multiple model systems [2]. The major cell type harboring latent KSHV infection in vivo are B cells. Latent infection of B-cells can result in KSHV-induced lymphoproliferation, which is linked to primary effusion lymphoma (PEL) and multicentric Castleman’s disease. KSHV-positive PEL-derived cell lines are a powerful source of material for studying the latent KSHV epigenome. However, constitutively infected cell lines are not ideal to investigate the early events in the establishment of latency in a non-transformed cell type. Despite being the major latent cell type, human B cells are largely refractory to de novo KSHV infection in vitro. Hence, to model KSHV latent or quiescent infection in vitro, many labs utilize endothelial-like cell lines, although the most commonly used cell line (SLK) appears to be a renal carcinoma cell line [89,117,118]. H3K27me3 has clearly been found to be enriched on silenced lytic promoters in PEL-derived lines and latent SLK cells, as well as following in vitro infection of endothelial cells and human PBMCs [85,89,90,117], and its enrichment closely mirrors that of the KSHV genome from Kaposi Sarcoma clinical isolates [86]. PRC2 components EZH2 and SUZ12 are detectable on lytic promoters marked by H3K27me3 [90,117], and inhibition or knockdown of EZH2 enhances viral gene expression in PEL-derived lines or following de novo infection of SLK cells [90,117]. H3K27me3 is lost upon reactivation induced by sodium butyrate (a histone deacetylase inhibitor) or the exogenous expression of a lytic transactivator, and over-expression of H3K27 demethylase JMJD3 increases viral gene expression and incidence of reactivation from in vitro models of latency in B-cells and SLK cells [89,90]. Together, data from these studies indicate a functional role for H3K27me3 in maintaining KSHV lytic gene silencing during latency.

KSHV is perhaps the best characterized of the herpesviruses in terms of the presence of PRC1 on the latent genome. Multiple studies have detected H2AK119ub1 on lytic promoters, with data indicating that it is deposited prior to K27 methylation in SLK cells [117]. The core PRC1 component RING1B is also present at lytic promoters enriched for H2AK119ub1 in addition to vPRC1-associated proteins KDM2B and RYBP [117,119,120]. Therefore, vPRC1 and H2AK119ub1 appear to be important in fHC formation on the latent KSHV genome. However, KDM2B depletion does not perturb RING1B enrichment on the KSHV genome, suggesting it might not recruit vPRC1, or that there are alternative, compensatory pathways to fHC formation [121]. Intriguingly, a recent study found that knockdown or inhibition of the PRC1 component PCGF4/Bmi1 in B cells resulted in loss of H2AK119ub1 from the Rta lytic gene promoter and induced increased levels of viral transcripts [122]. Therefore, interrogating the precise molecular dependence of KSHV latency on different components of PRC1 in different cell types will help elucidate cross talk between the different PRC1 complexes and their resulting effects on three-dimensional chromatin interactions and gene expression.

There is also evidence for fHC silencing in Epstein–Barr virus (EBV). Of note, EBV latency is classified into “types”, which are characterized by the differential expression of latent genes/proteins that allow the persistence and proliferation of the cell [123]. Type I latency, which is observed in EBV-positive Burkitt lymphoma cells and derived lines, express EBNA1, EBERs, and BamHI A rightward transcripts (BART) transcripts. Type II latency, which is observed in NK/T lymphoma cells, express LMP-1 and LMP-2 in addition to the Type I latency genes. Following the EBV infection of B cells in vitro, the B cells are transformed and form lymphoblastoid cell lines (LCLs), which can be passaged continuously. LCLs exhibit Type III latency, which additionally express EBNA-3s, EBNA-LP, and BHRF1 miRNAs. In vivo, EBV is thought to transition between latency programs as the infected B cell differentiates and gains access to the memory B cell pool [124]. Different latent promoter elements are activated or silenced depending on latency type, likely in part due to epigenetic regulation, as reviewed by Tempera and Lieberman [125].

H3K27me3 and EZH2 have been demonstrated on the promoter of ZEBRA/BZLF1 (an early gene required for lytic induction) in latency Type I, II, and III cells and on several lytic promoters in Type I cells [91,92,126]. The vPRC1 component KDM2B has been detected at EBV promoters and exerts a repressive effect, although whether this is linked to vPRC1 function and H2AK119ub1 is not yet known [127]. The exact contribution of H3K27me3 in silencing is also unknown. H3K27me3 is lost following lytic induction by some triggers, such as exogenous ZEBRA expression [126], but not others, including TPA (a protein kinase C activator), A23187 (a calcium ionophore), and sodium butyrate (a histone deacetylase inhibitor) [92]. This may be reflective of the stimulus used given that ZEBRA expression bypasses the process of re-expression of ZEBRA, the promoter of which is marked with H3K27me3 during latency, and likely recruits chromatin remodelers [128]. Other stimuli that act on signal transduction pathways (for example, TPA) result in an increase in transcriptional activation-associated histone modifications but no detectible decrease in H3K27me3 [92]. It is possible that the loss of H3K27me3 on just a few genomes, which would be difficult to detect, may be sufficient for lytic induction.

Further, EZH2 knockout, knockdown, or inhibition alone is not sufficient to induce robust changes in ZEBRA expression in Type I or Type III latent cells [92,129,130], although this does not necessarily imply they are not important at maintaining long-term repression as additional signaling events are likely required to induce transcriptional activation. In support of this, PRC2 inhibition has been shown to enhance viral gene expression when in combination with other lytic induction triggers in Type I and III cells [92,129]. Therefore, H3K27me3 appears to be important for maintaining silencing. Its inhibition alone promotes reactivation in response to additional triggers but does not directly result in entry into the lytic cycle.

## 4. Potential Mechanisms of Polycomb Group Protein Recruitment to Herpesvirus Genomes

The specific recognition mechanisms by which PcG proteins target herpesvirus genomes have not been elucidated for the majority of herpesviruses. De novo targeting of PcG-mediated fHC has been problematic to study, even outside of the context of viral infection, and there are still many questions remaining concerning the mechanisms that target polycomb group proteins to regions of DNA. Additionally, there is evidence that the mechanisms of targeting likely vary between cell types and different regions of fHC, so a one size fits all model of fHC may not apply. Some of the most in-depth studies on fHC targeting come from studying development, where there are multiple forms of fHC establishment, including: de novo chromatinization of the paternal genome after fertilization; X chromosome inactivation in female embryos along with silencing of imprinted regions; and developmentally regulated genes, along with the collapse (via SUZ12/EED depletion or EED inactivation) and subsequent re-initiation of regions of H3K27me3 [11,12]. From these studies, a number of key concepts have emerged including the kinetics of H3K27 methylation, the role of ncRNAs, the requirements for H2A ubiquitination, the involvement of polycomb accessory proteins and the contribution of fHC to gene silencing. Below, we synergize these concepts with what is currently known about potential mechanisms of PcG-mediated fHC targeting to latent herpesvirus genomes. The polycomb repressive core components, associated accessory proteins, and fHC marks that have been demonstrated on latent herpesvirus genomes, as well as our proposed mechanisms of how these components may facilitate fHC targeting, are summarized in Figure 3.

### 4.1. Kinetics of H3K27me1/2/3 Targeting to Herpesvirus Genomes

H3K27me3 is considered the hallmark of PcG-mediated fHC and the most well characterized fHC-associated modification on latent herpesvirus genomes. One potential model for H3K27me3 targeting is that it occurs on all incoming viral genomes irrespective of cell type, but during a successful lytic replication cycle the virus is able to overcome this mechanism of gene silencing. If H3K27me3 acts as a host defense mechanism to limit incoming viral gene expression then it would be rapidly deposited onto incoming viral genomes and persist through latency. However, in experimental model systems where deposition of de novo H3K27 methylation to regions of cellular DNA has been tracked over time, a theme has emerged that trimethylation of K27 takes approximately 36 hours to form and spread [11,12]. In the case of a rapidly replicating virus like HSV-1, which can produce progeny virus by 12–18 h post-infection, this slow deposition of H3K27me3 is unlikely to profoundly affect viral gene expression. However, progression to full trimethylation need not occur for fHC to contribute to herpesvirus gene silencing. When de novo K27 methylation is tracked over time, H2K27me1 and K27me2 occur rapidly, followed later by H3K27me3 formation and spread [11].

In the Gammaherpesviruses, H3K27me3 likely does not accumulate on genomes destined to become latent until an initial round of viral gene expression has been completed. A “pre-latent” phase has been proposed in de novo EBV and KSHV infection systems [74,117,131,132,133]. Specifically, for KSHV, H3K27me1 levels rise and then decrease concurrently with the rise of H3K27me3 beginning around 24 h post-infection [117]. This suggests a transition from H3K27me1 to me2 to me3, although H3K27me2 has remained unexplored in this system. Indeed, the association of H3K27me2 and consequence for herpesvirus gene silencing has been under-studied. Intriguingly, H3K27me2 is the more abundant modification on the host genome, although its direct role in gene silencing compared to trimethylation is not understood [21]. A recent study did identify a novel histone reader specifically for H3K27me2 as the tudor domain-containing protein PHD finger protein 20 (PHD20) [134]. Binding of PHD20 to H3K27me2 results in recruitment of the repressive Mi-2/nucleosome remodeling and deacetylase complex, at least in cancer cells, providing a direct mechanistic link between H3K27me2 and transcriptional repression.

A further consideration is the nucleosomal structure of incoming herpesvirus genomes. Upon initial infection the viral genomes lack histones and are non-nucleosomal [67,68,69,70,71,72]. Multiple lines of evidence point to PRC2 activity being enhanced by a dense, polysomal structure with an optimal DNA linker length of 40 bp [135]. This may relate to initial binding of the complex to a nucleosome already containing H3K27me3 via the EED aromatic cage, which, based on data obtained from cryoEM of AEBP2-containing PRC2, results in the positioning of the EZH2 SET domain in close proximity to the neighboring unmodified histone [30]. Hence, a dense nucleosomal structure likely facilitates the spread of K27 methylation from an initiating site. However, how nucleosomal density may co-ordinate de novo lysine methylation is currently unknown, and whether di- or tri- methylation could occur on isolated nucleosomes prior to formation of a dense nucleosomal structure on incoming herpesvirus genomes is unclear. In the case of HSV-1 and KSHV, H3 nucleosome occupancy accumulates prior to H3K27me3 [95,117]. For HCMV, H3 and H3K27me3 have not been directly compared; however, it has been demonstrated that H3 deposition occurs rapidly on the HCMV genome during latent infection, suggesting it may be deposited prior to H3K27me3 [136].

### 4.2. KDM2B/vPRC1-Mediated Recognition of Unmethylated CpGs

A discussed above, there is substantial evidence that fHC containing H2AK119ub1 occurs via vPRC1 recruitment [48,49], via association with different accessory proteins, one being KDM2B. KDM2B binds to unmethylated CpGs and results in the recruitment of the vPRC1 complex to ubiquitinate H2A [18,44]. H2AK119ub1 is subsequently recognized by PRC2.2 via JARID2, which has H2AK119ub1 reader activity [27]. Many of the human herpesviruses exhibit high genomic CG content and do not demonstrate detectable DNA methylation at the time of latency establishment. The KSHV genome exhibits an average GC content of 54%, raising the possibility of a role for KDM2B in polycomb-mediated recruitment [120]. In addition, data on enrichment of H2AK119ub1 prior to H3K27me3 hinted at this potential mechanism [117,120]. Accordingly, in a recent elegant study by Gunther et al., the authors detected KDM2B and H2AK119ub1 on KSHV episomes following de novo infected SLK cells [120]. H2AK119ub1 preceded recruitment of EZH2 and H3K27me3 by approximately 24–28 h, although H3K27me2 levels were not analyzed. An earlier study by Toth also detected non-canonical PRC1 component RYBP1 on the KSHV genome following infection of the same cell type [117]. In support of the model of KDM2B-mediated recruitment of vPRC1 to GC-rich regions followed by H3K27me3 deposition, a comparative epigenome analysis of KSHV and the related murid Gammaherpesvirus MHV-68 (which differ in terms GC-richness) showed that MHV-68 is largely devoid of CpG islands [120]. The latent KSHV genome exhibits profoundly higher levels of H3K27me3 than MHV-68, even upon infection of the same cell types, which is likely attributed to the higher GC-content of KSHV. Intriguingly, KDM2B depletion during the establishment of latency results in increased KSHV lytic gene expression and a concomitant increase in H3K36me3 and H3K4me3 [121]. The repressive effects of KDM2B on KSHV lytic gene transcription occur prior to the observed increase in H3K27me3, and KDM2B depletion did not alter RING1B binding to the viral genome, but an alternate mechanism by which KDM2B may represses lytic gene expression may exist. This proposed mechanism of fHC targeting to the KSHV genome, in which vPRC1 recruitment mediated through GC-rich motifs is followed by PRC2.2 recruitment and H3K27me3 deposition, is represented in Figure 3.

The known mechanism of recognition of H2AK119ub1 by PRC2.2 in ESCs is via JARID2. However, in many lineage-committed cells, including primary lymphocytes and keratinocytes, the predominant isoform of JARID2 is ΔN-JARID2 [37], which lacks PRC2 binding ability and acts to promote transcription of lineage-specific genes. Therefore, given the predominant lack of the full length JARID2 in primary, lineage-committed cells, the exact mechanism for PRC2.2 recruitment to H2AK119ub1-associated promoters and requirement for JARID2 remains to be determined. Interestingly, EBV and KSHV both express miRNAs that downregulate JARID2, and this is hypothesized to lead to the loss of tumor suppression [137,138]. Because of the central role JARID2 in vPRC1/PRC2.2-mediated PcG-silencing, the observation of H2AK119ub1 on herpesvirus genomes, and the potential change in function following differentiation, examining the role of JARID2 in herpesvirus latency will uncover how PRC2.2 contributes to gene silencing outside of the context of embryonic stem cells.

### 4.3. Sequence-Specific Polycomb Recruitment

In *Drosophila*, Polycomb complexes are recruited to specific sequences known as Polycomb response elements (PREs). Mammals lack this direct sequence-specific recruitment of polycomb group proteins. However, by association with accessory proteins or transcription factors, there is evidence of polycomb complex targeting to specific promoters for gene repression. Proteins identified thus far that may mediate sequence specific polycomb recruitment include MGA, E2F6, RUNX1 and REST [139,140,141]. One intriguing protein that is potentially linked to herpesvirus latency is RUNX1. RUNX1 has been shown to be involved in Ring1b recruitment, and subsequent H2AK119ub1 to RUNX1 on the cellular genome binding sites in primary thymocytes [140]. RUNX1 binding sites are over-represented in HSV-1 and HSV-2, particularly in regions surrounding transcriptional start sites of lytic promoters [142]. Further, RUNX1 overexpression has been found to repress HSV gene expression during lytic infection. Together, this evidence suggests RUNX1 acts as a repressor of HSV gene expression, although the function of RUNX1 in HSV latency and connection to Polycomb silencing of HSV lytic genes remains to be determined. Of interest, RUNX1 plays a role in the development of neurons and cells of myeloid and lymphoid lineages, sites in which human herpesviruses establish latency. It is notable that EBV infection downregulates RUNX1, which is believed to relieve RUNX1-mediated growth repression and contribute to EBV’s lymphoproliferative effect [143]. Whether RUNX1 plays a role in directly modulating Polycomb recruitment to the EBV genome is not known.

A second transcriptional repressor, RE1-silencing transcription factor (REST, also known as neuron-restrictive silencing factor (NRSF)), is linked to the silencing of neuronal genes in non-neuronal cells. REST can interact with PRC1 and PRC2 and maintains PRC1/2 binding to neuronal genes in the human teratocarcinoma NT2-D1 cells and mESCs [141]. However, it should be noted that the role of REST in Polycomb recruit is complex and context-dependent as REST can also limit PRC2 binding around CpG-rich regions in murine embryonic cells [141]. The HSV-1 genome carries numerous predicted REST binding sites, and insertion of a dnREST, which retains its DNA-binding domain but lacks the regions that are responsible for interacting with repressors, into the HSV genome was found to disrupt latency in comparison to insertion of the WT REST [144]. In addition, a recombinant HSV-1 with a wild type REST insertion failed to reactivate in a TG explant model even in the presence of HDAC inhibitors [145]. This suggests that REST exerts a repressive effect on the viral genome through association with repressors other than histone deacetylases, although, in this study, expression of the LAT and viral miRNA was reduced following infection with the wild-type REST virus. Therefore, a direct contribution of REST to polycomb silencing on the HSV genome remains to be determined.

### 4.4. Regulation by RNA

RNA has emerged as playing an important role in the modification of fHC and known cellular mechanisms are summarized in Figure 4. Multiple proteins that make up core PRC complexes in addition to accessory proteins have RNA binding capabilities, including EZH2, Suz12, hnRNPK and ATRX. A recent study demonstrated that RNA itself and the RNA binding domain of EZH2 is required for PRC2 binding to chromatin [146]. Hence, RNA plays a key role in PRC2 recruitment and Polycomb silencing. However, the relationship between Polycomb recruitment and RNA binding is far from simple, and although Polycomb complexes can be recruited via binding to RNA, the methyltransferase activity of PRC2 is inhibited by RNA binding and single-stranded RNA competes for PRC2 DNA binding [147] (Figure 4B). In some instances, non-coding RNAs can recruit Polycomb complexes for the initiation of silencing, while RNA can also modulate the activity of PRC2. Therefore, a theme is emerging in which polycomb can be recruited via RNA interactions but additional factors such as pre-existing histone PTMs, presence of transcription factors, composition of the polycomb complexes, nature of the RNA and active transcription ultimately regulate whether fHC is formed [148]. Given that multiple herpesviruses express long non-coding (lnc) RNAs during latency, it is likely that they also modulate the function and/or recruitment of Polycomb proteins. However, like cellular non-coding RNAs, these mechanisms are likely complex and dependent on additional factors and/or the pre-existing chromatin template.

In support of a role for RNA in the establishment of fHC, one of the most well characterized non-coding RNAs to participate in silencing is the X-inactive-specific transcript (Xist) [51]. Xist is 17 kb RNA that is necessary for X chromosome inactivation, a process that regulates dosage compensation in female XX cells. Not only is Xist necessary for X chromosome inactivation in ES cells, it can also induce autosomal gene silencing as an exogenously inserted transgene. Distinct domains of the Xist RNA interact with different repressive proteins to orchestrate multiple events in epigenetic silencing of the X chromosome. A region of the Xist (the B repeat), recruits vPRC1 to catalyze H2AK119ub1 and permit recruitment of PRC2.2 via JARID2 [43,46]. The Xist does not bind to PRC1 directly but instead recruits vPRC1 to sequences of the X chromosome destined for silencing via binding to hnRNPK [46] (Figure 4A). Xist also does not interact directly with chromatin; it is instead thought to increase the concentration of hnRNPK/vPRC1 in the vicinity of the region to be silenced. This is likely a common mechanism used by lncRNAs, at least in embryonic stem cells, where additional RNAs also involved in imprinting (Airn, Kcnq1ot1 and Meg1) interact with hnRNPK and Polycomb proteins [149]. It should be noted that these lncRNAs also participate in gene silencing in a non-polycomb-dependent manner, for example, by potential transcriptional interference of anti-sense transcripts (notably Airn and Kcnq1ot1) and recruitment of additional silencing proteins including histone deacetylase and demethylases (reviewed in [149]).

In addition to silencing in *cis*, additional lncRNAs have been discovered that silence distal regions of host chromatin. HOTAIR was the first lncRNA to be identified that worked silenced regions of the cellular genome in *trans*. At the time of its discovery, PRC2 was thought to initiate Polycomb silencing prior to PRC1 recruitment [149]. This result was similar to the model proposed for Xist-mediated recruitment of PRC2 in *cis* via binding to a region of Xist (RepA) as well as to the anti-sense transcript via a two-hairpin structure to PRC2 and, therefore, PRC2 was thought to bind to similar stem-loop structures [150]. However, further studies have since found that PRC2 (and in particular EZH2) has an affinity for a wide range of RNA species, with the highest binding affinity for RNA containing G tracts folded into G quadruplexes ([G3-5N1-5]4-6; known as G4 structures) [147,151]. EZH2 also binds with a moderate affinity for G-track motifs, as well as hairpin structures, albeit with lower affinity than G-track motifs [51]. Interestingly, a recent study addressed the conundrum as to how lncRNAs play a role in promoting PRC2 recruitment and H3K27me3 formation when the complex is known to be inhibited by RNA binding. It was found that although binding to ssRNA inhibited PRC2 activity, binding of dsRNA can relieve this repression [152]. Hence, this study helps resolve how HOTAIR potentially promotes polycomb silencing by base paring with nascent RNA transcripts that have homologous sequences and, thus, overcome the repression of ssRNA imposed by binding to EZH2.

As yet, no herpesvirus non-coding RNAs have been identified that directly recruit Polycomb group complexes to viral genomes during latency akin to Xist. However, it is possible that herpesvirus non-coding RNAs do regulate the nature of fHC on viral genomes. Herpesvirus long non-coding RNAs expressed in latency include the latency-associated transcript (LAT) of HSV, lncRNA 4.9 of HCMV, and the BARTS and EBERS of EBV [104]. Some of these are incredibly abundant in the latently infected nucleus: The HSV LAT intron is present at more than 10^4^ copies per cell and the EBV EBERs at about 10^5^–10^6^ [104]. The KSHV PAN RNA, robustly expressed during the transition to lytic replication, is present at >10^5^ copies/cell [104]. Given the complexities in the nature of fHC, different routes to silencing and potential modulation in the recruitment and activity of PRC2 complexes, it is possible that herpesvirus non-coding RNAs could modulate the nature of the PRC1 or PRC2 proteins recruited, the type of fHC silencing and the genomic locations of fHC on either the cellular or host genomes.

There are hints in the literature that herpesvirus lncRNAs modulate the fHC on latent viral genomes. For HSV-1, the most abundant transcript present in latently infected neurons is the LAT. The LAT consists of a primary 8.3 kb transcript, which is spliced into major 1.5 and 2 kb LAT introns. The LAT and smaller RNAs processed within it have been implicated in the silencing of viral expression during acute and latent infection [103,153,154,155]. During acute infection of murine ganglia, where both the LAT and lytic transcripts are abundantly expressed, the LAT has been found to reduce lytic gene transcription [155]. During latency, expression of at least two viral lytic transcripts is increased in a LAT mutant virus [155]. In addition, experiments using LAT+/− viruses with a HCMV-MIEP luciferase cassette inserted into a lytic region of the genome found decreased MIEP repression in the LAT-null virus [103]. Whether these changes in gene expression relate to differences in fHC silencing, however, remains controversial, as one study found decreased association of H3K27me3 in the absence of the LAT using multiple virus mutants [82], whereas a second study found increased H3K27me3 [84]. However, it should be noted that the second study was carried out with a LAT mutant that has since been sequenced and found to have multiple genetic changes that confound the interpretation of the data directly relating to the function of the LAT, including regions substituted with HSV-2 sequence and altered amino acid sequences within 13 different ORFs [156]. A third study found a trend towards a decreased—but not a significantly changed—level of H3K27me3 without the LAT [103]. One problem with assessing the contribution of the LAT, however, is possible additional functions, including potentially impacting neuronal survival [157,158], that could alter the pool of latently infected neurons in vivo. However, together these data indicate that the LAT could impact the nature of fHC on the latent genome, but not be fully required for targeting. In support of this, SUZ12 appears to be recruited to similar levels to the lytic viral promoters tested even in the absence of the LAT during the establishment of latency [95].

As of yet, no specific interactions between the HSV LAT and polycomb group complexes or associated proteins have been identified. As previously mentioned, the PRC2 complex in particular has promiscuous RNA binding activity and lncRNAs expressed by other human herpesviruses have been found to interact with the complex. As previously mentioned, PRC2 interacts with the highest affinity to G-quadruplexes. Intriguingly G-quadruplexes have been demonstrated on the immediate early promoters/regulatory regions of HSV-1, HSV-2, VZV, HMCV, EBV, and KSHV, although the trend was less obvious for EBV (Figure 4D) [159,160,161]. However, how these interactions modulate latency and/or reactivation are not clear. The HCMV lncRNA 4.9 may regulate PRC2 function at a lytic promoter. During a latent infection of CD14+ monocytes, lncRNA 4.9 has been demonstrated to interact with PRC2 components EZH2 and SUZ12, and the RNA itself is enriched at the MIEP, which is also enriched for H3K27me3 [98]. These combined data suggest that lncRNA 4.9 may play a role in modulating fHC formation on the latent HCMV genome, potentially akin to the role of HOTAIR in forming a dsRNA structure and promoting PRC2 enzymatic activity (Figure 4C). However, whether there is any homology between regions of lncRNA 4.9 that are important for silencing, and MIEP-derived RNA, remains to be determined.

Conversely, herpesvirus lncRNAs could inhibit PRC2 function by binding as a ssRNA, and it is feasible that abundantly expressed transcripts such as the LAT and RNA 4.7 have both positive and negative effects on PRC2 binding. lncRNA expression during reactivation could also modulate PRC2 recruitment away from DNA and bind as ssRNA to inhibit the enzymatic activity. KSHV PAN RNA is robustly expressed following lytic induction. PAN RNA has been proposed to play a role in lytic gene induction as it is broadly detectable across the viral genome and its absence is linked to the inhibition of reactivation [162]. PAN RNA has been proposed to interact with SUZ12 and EZH2 [162] and may potentially divert PRC2 away from chromatin and/or restrict PRC2 function, although this remains to be tested. It is particularly interesting that the PAN RNA is not associated with chromatin, which does not negate a role for interacting with PRC2, especially if the PAN RNA displaces PRC2 from chromatin [163]. However, additional studies have suggested that PAN does not regulate gene expression during reactivation [163]. It is unknown whether PAN may still modulate transcription following a physiological reactivation stimulus (this study used valproic acid, a histone deacetylase inhibitor). Therefore, it remains to be resolved whether PAN RNA can modulate Polycomb binding during KSHV reactivation and whether this depends on how reactivation is induced.

## 5. Future Directions

As discussed here, there are many unknowns regarding the formation of fHC on latent herpesvirus genomes, the exact nature of the polycomb group proteins present and the potential mechanisms of de-repression for reactivation. There is much still to be uncovered that will both inform on how polycomb silencing is targeting de novo in distinct cell types and on the mechanisms of herpesvirus gene silencing. In addition to understanding the mechanisms of fHC formation on viral genomes, the ability to track herpesvirus latency and reactivation in distinct cell types provides novel model systems to understand long-term effects of different fHC structures on chromatin silencing.

### 5.1. Contribution of Cell-Type Specific PRC Complexes

Much of the data on the mechanisms of Polycomb silencing come from studies carried out in pluripotent cells or upon early stages of differentiation. Herpesviruses establish latency in distinct cell types for which the exact composition of Polycomb complexes has yet to be resolved. A general theme in the literature is that the mechanisms of polycomb silencing become more fixed as cells become more differentiated; for example, the switch from vPRC1/H2AK119ub1 mediated silencing [57], the down regulation of EZH2 and EED as cells become more differentiated, and the replacement of JARID2 with a cleaved form that is unable to recruit PRC2 to H2AK119ub1 [37]. Whether these changes occur in cell types latently infected with human herpesviruses and the impact on incoming genomes remains to be resolved. The majority of the current literature on PRC2 recruitment centers around EZH2. However, as cells become more differentiated, they switch to reduced EZH2 expression and higher levels of EZH1. Even post-differentiation as cells undergo maturation, the expression patterns of Polycomb group proteins change. An example being neurons that undergo an intense maturation period in the post-natal to adult period, during which time EZH2 and EED are down-regulated, and EZH1 levels remain unchanged [102]. There is evidence that PRC2-EZH1 is functionally distinct from PRC2-EZH2. PRC2-EZH2 displays allosteric activation, whereas EZH1 activity is unresponsive to activation by H3K27me3 [35]. PRC2-EZH1 activity is inhibited by free DNA whereas PRC2-EZH2 is unaffected by non-nucleosomal DNA [35]. This may explain why PRC2-EZH2 is the major methyltransferase in ES cells as the chromatin structure is more open and higher levels of H3K27me3 favor allosteric activation. In contrast, the more compact chromatin structure in differentiated cells would result in less inhibition of EZH1. However, herpesvirus genomes start out as a non-nucleosomal structure, the presence of non-nucleosomal genomes would likely initially inhibit the activity of EZH1, although it is possible that association with different PRC2 co-factors may alter the activity of EZH1. To our knowledge, the contribution of EZH1 and associated proteins to herpesvirus latency has yet to be resolved.

### 5.2. Contribution of Different PRC Components to de Novo Silencing and Long-Term Repression

Tracking the initial silencing of herpesvirus lytic gene expression and prolonged silencing through latency serve as excellent systems to understand the contributions of different Polycomb complexes, their associated proteins and enzymatic activity to either de novo gene silencing or maintained repression. Key questions that remain unanswered include the role for H2AK119ub, H3K27me2 and H3K27me3 in either switching off lytic gene expression or their targeting to genes that are already transcriptionally silent. In addition to the histone post-translation modifications, what role do the associated histone readers play in gene silencing? Is silencing maintained via H2AK119ub1 to prevent transcriptional elongation, via chromatin compaction or does Polycomb binding regulate three-dimensional interactions within viral genomes themselves or to regions of host chromatin? Answering these questions will provide mechanistic insight into long term persistence of latent herpesvirus genomes and is also required to understand how Polycomb silencing could be reversed to permit reactivation following an appropriate stimulus.

### 5.3. Viral Manipulation of the Polycomb Structure of the Virus and Heterogeneity in Latency

Given the high degree of heterogeneity of Polycomb silencing, it is possible that herpesviruses actively manipulate the nature of fHC on the viral genome to a form most advantages for viral persistence and/or reactivation. Polycomb silencing has varying degrees of silencing and levels of compaction depending on the associated proteins. Although there is evidence of redundancy between different components of PRC2.1 and PRC2.2, whether there is true redundancy in terms of long-term gene silencing, the nature of PRC1 recruited and ability to convert back to euchromatin for reactivation is not known. The nature of PRC1 recruited would ultimately impact the epigenetic structure of viral genomes. Chromatin compaction is carried out by CBX2, 4 and 6 but CBX7 lacks this ability [10]. CBX2 is the only known PRC1-asscoiated CBX protein to participate in Phase separation [56]. Understanding the nature of the proteins associated with viral genomes (as mentioned above) in addition to whether the viruses themselves drive one form of silencing over another may inform on what is the most advantageous form of silencing most amenable to reactivation. In addition, herpesvirus latency is heterogeneous in nature and often only sub-populations of viral genomes reactivate at any one time. A key question is whether the heterogeneity related to the nature of the viral chromatin at the time of reactivation? Although speculative at this point, understanding if this is the case may help to aid the development of novel therapeutics that drive herpesvirus latency into its most silent form that is refractory to reactivation.

### 5.4. Mechanisms of Reversal for Reactivation

Herpesvirus reactivation can be induced experimentally and tracked over time, and therefore provide model systems to investigate how Polycomb silencing can be overcome to result in gene expression. One potential mechanism is via recruitment of histone demethylases (to remove K27 methylation) and/or deubiquitinase. A role for K27 demethylases has been found for full HSV [97,105,164], HCMV [111] and KSHV [89,90]. However, the mechanism of recruitment remains unknown. The contribution of Polycomb repressive deubiquitinase (PR-DUB) in herpesvirus reactivation has yet to be explored. Given the link between H2AK119ub1 and inhibition of transcriptional elongation, exploring the differential requirements for H2AK119ub1 removal versus H3K27me2/3 removal will be an interesting avenue to explore.

There also remains the possibility of Polycomb eviction for reactivation, which may occur independently of histone demethylase activity. The initial induction of gene expression during HSV reactivation has been shown to be independent of the activities of the K27 histone demethylases [97,164]. However, the mechanism by which gene expression is able to initiate under these conditions is not known. One proposed mechanism is via phosphorylation of the S28 residue, which occludes binding of PRC2 to H3K27me3 and permits gene expression [165]. This combined histone H3K27/S28 methyl/phospho switch has been observed on cellular promoters following cell stress [165,166] and in neurons following stimulation [167]. A similar mechanism for potentially overcoming H3K9me3 has been observed on the HSV genome in reactivation [97,164]. However, whether H3S28p is linked to reactivation of any of the human herpesviruses is as-yet unknown. A further mechanism with the potential for quick action following a reactivation stimulus is RNA expression. G quadruplex RNAs have been shown to compete for PRC2 binding, leading to reduced occupancy and H3K27me3 [148,151], and G quadruplexes are prevalent on herpesvirus genomes [161]. Finally, a possibility that is not mutually exclusive is via the recruitment of pioneer factors, transcription factors that can access polycomb repressed chromatin and recruit chromatin remodeling complexes. In the case of EBV, pioneer factor BZLF1 is described as the switch between latency and lytic replication, recruiting remodeling complexes that can remove H3K27me3 [74].

As with the diverse processes explored for the establishment of fHC, overcoming H3K27me3-mediated repression most likely differs by the cell differentiation state, cell type-specific transcript and protein expression, reactivation trigger and other factors specific to each herpesvirus. Understanding the full nature of PcG silencing on latent herpesvirus will also be required to determine how silencing is reversed for reactivation. Ultimately, understanding the mechanisms PcG silencing and how it is reversed for reactivation may ultimately enable the development of therapeutics that prevent disease associated with herpesvirus recrudescence.

## Figures and Tables

**Figure 1 viruses-13-01470-f001:**
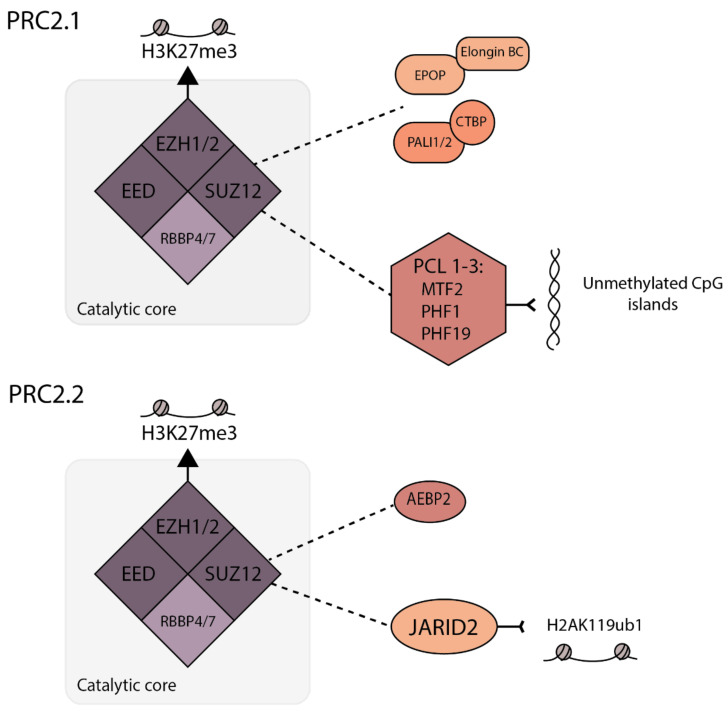
The composition of PRC2.1 and PRC2.2 complexes. Polycomb repressive complex 2 (PRC2) has a catalytic core of EED, EZH1/2 and SUZ12 along with RBBP4/7. PRC2 carries out all three methylation states of H3K27, but H3K27me3 is the only one stably bound by the complex. To form the PRC2.1 complex, the core can interact with either PALI1/2 and CTBP or EPOP and Elongin BC. One pair excludes the other from joining the complex. SUZ12 in PRC2.1 also interacts with one of three PCL proteins, but this interaction does not compete for the aforementioned interacting pairs. PCL proteins enable binding to unmethylated CpG islands. PRC2.2 forms by SUZ12 in the same catalytic core interacting with JARID2 and AEBP2. Unlike PRC2.1, JARID2 enables PRC2.2 recruitment to sites of H2AK119ub1.

**Figure 2 viruses-13-01470-f002:**
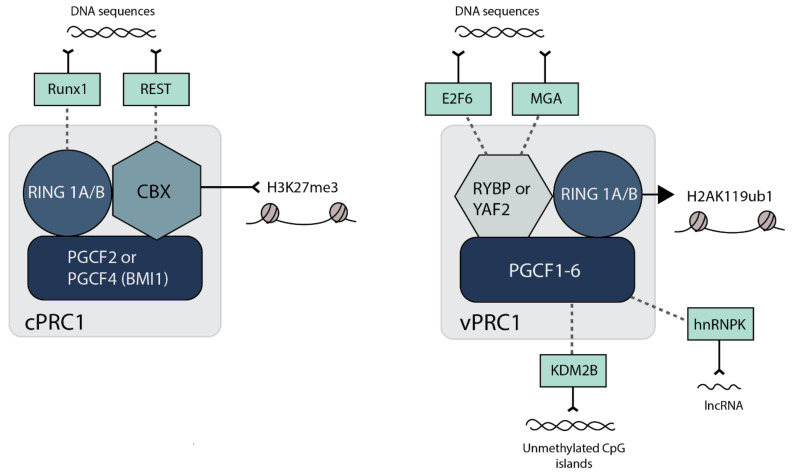
The components of canonical and variant PRC1 complexes and interacting transcription factors. Canonical PRC1 (cPRC1) consists of one of two PGCF proteins (PGCF2 or PGCF4), along with RING1A/B and one of five chromobox (CBX) proteins. Although catalytically active, cPRC1 is thought not to contribute to the majority of H2AK119ub1. CBX proteins directly bind to H3K27me3, so cPRC1 recruitment to chromatin is dependent on PRC2 H3K27 methylation activity. Variant PRC1 (vPRC1) contains any of the six PGCF proteins, RING1A/B, and RYBP or YAF2. RYBP/YAF2 do not bind H3K27me3 and vPRC1 recruitment is thus independent of PRC2 activity. RYBP stimulates RING1A/B activity, and vPRC1 writes H2AK119ub1. Transcription factors RUNX1 and REST can interact with cPRC1 via RING1B and CBX7/8, respectively, recruiting the complex to specific target sequences of DNA. E2F6 and MGA can similarly recruit vPRC1 to specific DNA sequences by interaction with RYBP/YAF2, while hnRNPK can do so through interaction with PGCF3/5 and long-noncoding lncRNA. KDM2B can recruit vPRC1 to non-sequence specific, unmethylated CpG islands.

**Figure 3 viruses-13-01470-f003:**
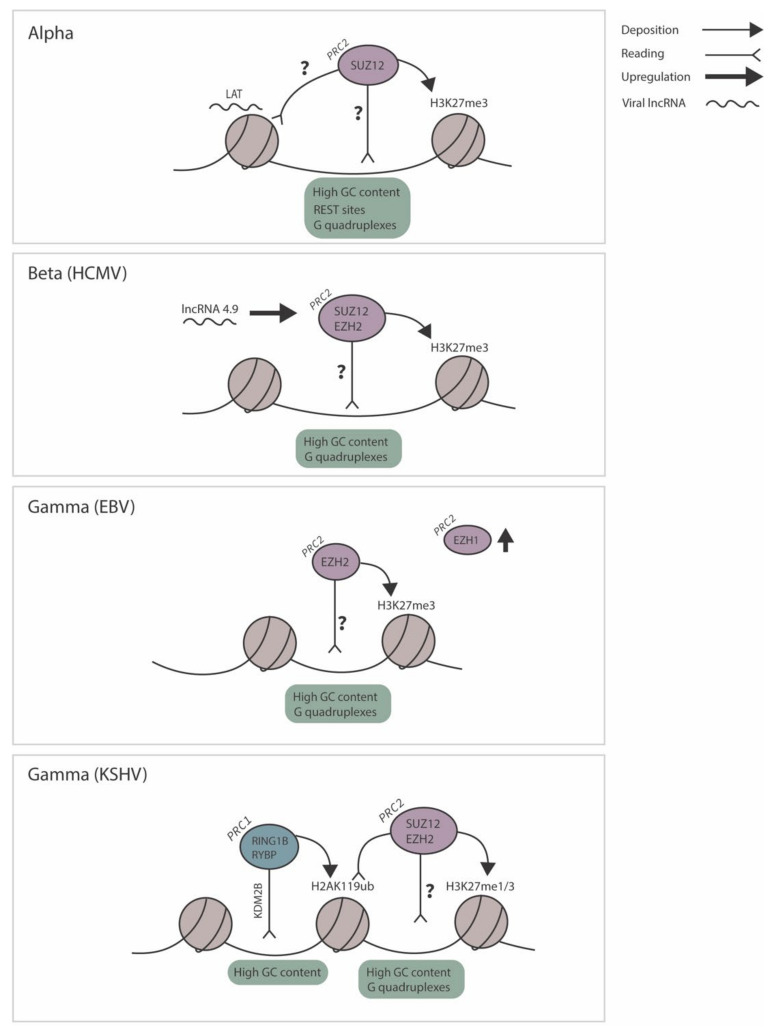
Proposed mechanisms of facultative heterochromatin targeting are demonstrated for representative herpesviruses from each family. For Alpha-herpesvirus HSV-1, we propose that PRC2.1 may be recruited to the genome through intrinsic DNA elements, such as high GC content, REST sites, or G quadruplexes, which are enriched on the genome. The LAT has also been demonstrated to play a role in H3K27me3 deposition, although independently of SUZ12 recruitment. In the case of Beta-herpesvirus HCMV, we propose that lncRNA 4.9 enhances the activity of the PRC2 complex, with which it has been demonstrated to associate. PRC2 may be recruited to the genome through similar over-represented intrinsic DNA elements. In the case of Gammaherpesvirus EBV, PRC2.1 may be recruited to the genome through G quadruplexes or high GC content. In the case of Gammaherpesvirus KSHV, we propose the recruitment of vPRC1 to the genome through GC content, and the subsequent deposition of H2AK119ub. PRC2, perhaps simultaneously, or on independent genomes, is also recruited to the viral genome to deposit H3K27me1/3.

**Figure 4 viruses-13-01470-f004:**
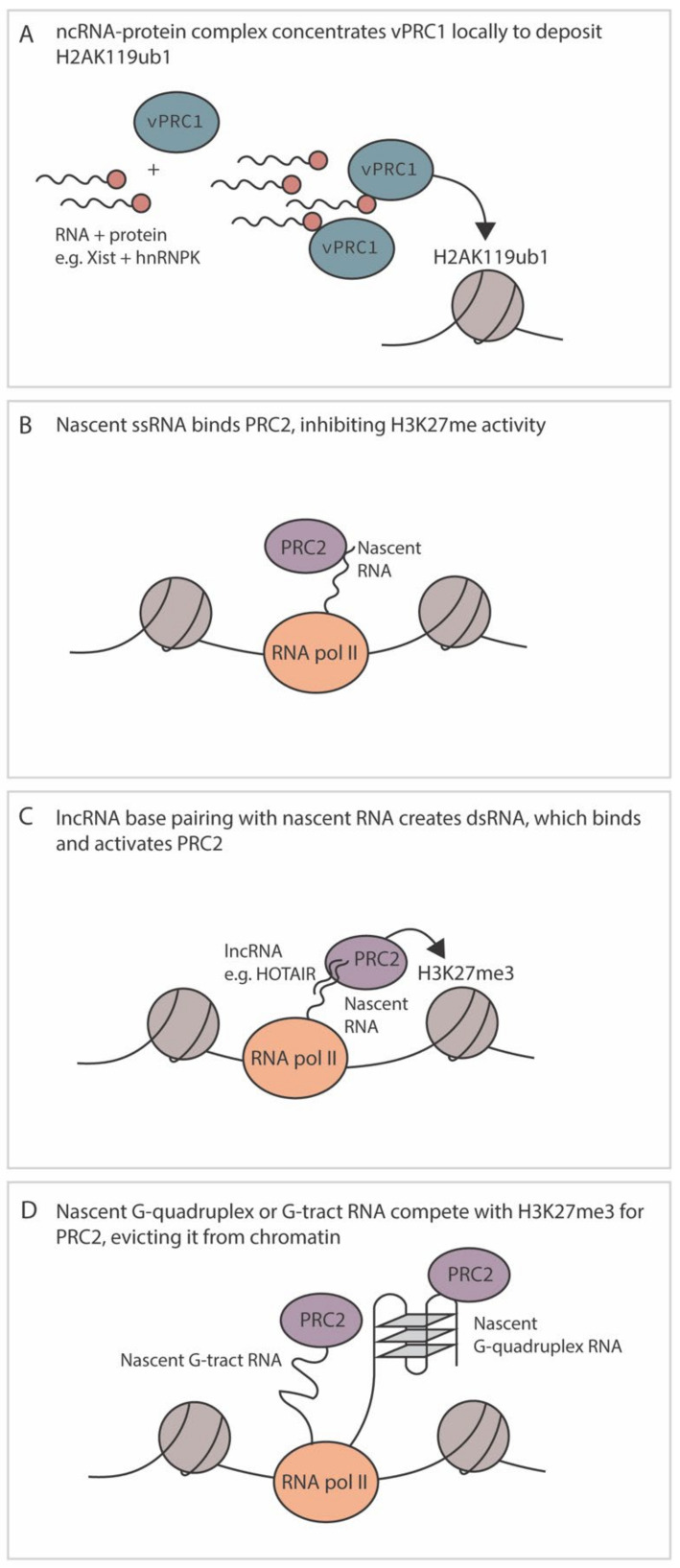
Cellular roles for RNA in modulating polycomb complex binding and facultative heterochromatin targeting. (**A**): Non-coding RNA paired with an RNA-binding protein can concentrate variant PRC1 near chromatin and promote subsequent H2AK119ub1 deposition. This has been demonstrated for Xist and hnRNPK. (**B**,**C**): Nascent RNA being transcribed by RNA polymerase II can inhibit PRC2 activity, but dsRNA formation by lncRNA base pairing to the nascent RNA can activate PRC2. HOTAIR is one such lncRNA. (**D**): Nascent G-tract or G-quadruplex RNA have high binding affinity for PRC2, and can thus compete with H3K27me3 for PRC2 binding. PRC2 is evicted from the chromatin as a result of this competition.

**Table 1 viruses-13-01470-t001:** Composition and abbreviations used for polycomb-associated proteins.

Complex	Full-Length Name	Abbreviation
**Polycomb repressive complex 1**		**PRC1**
	Ring Finger Protein 1A, B	RING1A, B
	Polycomb group RING finger protein	PCGF
**Canonical PRC1**		**cPRC1**
	Polycomb group RING finger protein 2, melanoma nuclear protein 18	PCGF2/MEL18
	Polycomb group RING finger protein 4	PCGF4/BMI1
	Chromobox 2, 4, 6, 7, 8	CBX2, 4, 6, 7, 8
	* Runt-related transcription factor 1	RUNX1
	* RE1-silencing transcription factor/ neuron-restrictive silencing factor	REST/NRSF
**Non-canonical/variant PRC1**		**vPRC1**
	RING1 and YY1-binding protein	RYBP
	YY1-associated factor 2	YAF2
	Lysine (K)-specific demethylase 2B	KDM2B
	Polycomb group RING finger protein 1–6	PCGF 1–6
	* E2F transcription factor 6	E2F6
	* MAX gene-associated protein	MGA
	* Heterogeneous nuclear ribonucleoprotein K	hnRNPK
**Polycomb Repressive complex 2**		**PRC2**
	Enhancer of Zeste 1, 2	EZH1, 2
	Suppressor of Zeste 12	SUZ12
	Embryonic ectoderm development	EED
	Retinoblastoma-associated proteins 46	RbAp46/RBBP4
	Retinoblastoma-associated proteins 48	RbAp48/RBBP7
**PRC2.1**		
	Elongin BC	No abbreviation
	Elongin BC- and PRC2-associated Protein	EPOP
	PRC2-associated LCOR isoform1, 2 C-terminal binding protein	PAL1, 2 CTBP
	Polycomb-like protein 1/PHD finger protein 1	PCL1/PHF1
	Polycomb-like protein 2/metal response element binding transcription factor 2	PCL2/MTF2
	Polycomb-like protein 3/PHD finger protein 19	PCL3/PHF19
**PRC2.2**		
	Jumonji and AT-rich interaction domain 2	JARID2
	Adipocyte enhancer-binding protein 2	AEBP2

* Accessory proteins linked with recruitment.

## Data Availability

Not applicable.

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
