# Peer review of "De Novo Polycomb Recruitment: Lessons from Latent Herpesviruses"

_viruses, 2021, doi:10.3390/v13081470_

Round 1

Reviewer 1 Report

The manuscript by Dochnal et al. presents a comprehensive review on de novo polycomb recruitment to herpesvirus genomes which results in viral genome silencing and ultimately a latent infection. The polycomb repressive complex is a cellular mechanism to regulate gene expression and has a well established role in developmental regulation and imprinting gene. During the early stages of infection, newly introduced viral genomes are associated with cellular histones and with repressive heterochromatin it enters into latency.  The repressive mark on facultative heterochromatin can be exchanged for an active mark of euchromatin during lytic reactivation.

This is a very well written manuscript that tackles a large amount of published research on this topic. There are many strengths of this manuscript including a good overview of the cellular proteins that make up the polycomb group repressive complex, and the known differences between cellular and viral interaction with these factors. Additionally, the authors nicely address each of the herpesvirus subfamilies, alpha, beta and gamma. It should be noted that the authors do a good job balancing the known interactions with polycomb factors from each of the prototypic members. This is not an easy task because of the varying models of latency and specific requirements for each of these different viruses.

Overall, an excellent manuscript. I see no major issue or areas that need to be addressed. I really appreciate the "Future direction" section, it summarizes the questions that remain in the field and identifies areas of continued research.

Author Response

Thank you! We appreciate the reviewer's time and kind words!

Reviewer 2 Report

This is a very nice review authored by Drs Dochnal, Francois and Cliffe discussing polycomb complex formation in many experimental systems including mammals, insects and herpesviruses.  
The first subject is discussion of polycomb complex formation in eukaryotic the systems. Next, latency strategies ofseveral human herpesviruses outlined then the review turns attention to the role of polycomb protein repression  during latency.
Overall this review contains valuable current information on an exciting and important area of biology and pathology related to human herpesviruses and gene regulation.
The authors encouraged to consider modifying the manuscript as follows.
(1) There are dozens and dozens of abbreviations of proteins and modified gene products in the review. This makes reading the article more difficult for people not actively researching this area. Try simplifying the article and use much fewer abbreviations.
(2) Figures 1 and 2 contain enormously complex data. These figures appears to be relevant to cellular polycomb repression regulation. The question is; are all of these interactions relevant to viral latency?
(3) Where is the DNA? This reader spent much time to find the most important fact in Fig 1 and 2. Consider drastic modification of these figures. For axample, depict DNA as a double helix.

Author Response

We thank the reviewer for their time and careful reading of the manuscript. 

(1) There are dozens and dozens of abbreviations of proteins and modified gene products in the review. This makes reading the article more difficult for people not actively researching this area. Try simplifying the article and use much fewer abbreviations.

We have simplified some of the abbreviations. We have also added a table with all the abbreviations for components of the polycomb complexes. 

(2) Figures 1 and 2 contain enormously complex data. These figures appears to be relevant to cellular polycomb repression regulation. The question is; are all of these interactions relevant to viral latency?

We have modified the figures and now include proteins of more direct relevance to herpesvirus latency. 

(3) Where is the DNA? This reader spent much time to find the most important fact in Fig 1 and 2. Consider drastic modification of these figures. For axample, depict DNA as a double helix.

This has been modified to simplify and DNA as a double helix added.